# The impact of atypical early histories on pet or performer chimpanzees

Hani D. Freeman and Stephen R. Ross

Lester E. Fisher Center for the Study and Conservation of Apes, Lincoln Park Zoo, IL, USA

## ABSTRACT

It is widely accepted that an animal's early history, including but not limited to its rearing history, can have a profound impact on later behavior. In the case of captive animals, many studies have used categorical measures such as mother reared or human reared that do not account for both the influence of human *and* conspecific interaction. In order to account for the influence of both human and conspecific early exposure to later behavior, we collected 1385 h of data on 60 chimpanzees, of which 36 were former pets or performers, currently housed at accredited zoos or sanctuaries. We developed a unique metric, the Chimpanzee-Human Interaction (CHI) Index that represented a continuous measure of the proportion of human and chimpanzee exposure subjects experienced and here focused on their exposure during the first four years of life. We found that chimpanzees who experienced less exposure to other chimpanzees as infants showed a lower frequency of grooming and sexual behaviors later in life which can influence social dynamics within groups. We also found chimpanzees who experienced more exposure to other chimpanzees as infants showed a higher frequency of coprophagy, suggesting coprophagy could be a socially-learned behavior. These results help characterize some of the long-term effects borne by chimpanzees maintained as pets and performers and may help inform managers seeking to integrate these types of chimpanzees into larger social groups, as in zoos and sanctuaries. In addition, these results highlight the necessity of taking into account the time-weighted influence of human and conspecific interactions when assessing the impact that humans can have on animals living in captivity.

## INTRODUCTION

Early life experiences have a significant impact on the behavioral development or the way in which individuals, both human and non-human, learn to interact with their environment (e.g., humans, *Kagan, 1996*; *Fox & Henderson, 1999*; non-human primates, *Parker & Maestripieri, 2011*). Outcomes of early life experiences have uncovered a broad range of potential impacts including those falling in both the social and non-social realms. Non-social variables include the impact of the physical environment on development (rats, *Leshem & Schulkin, 2012*), as well as physiological factors such as genetics (primates *Barr et al., 2003*; *Suomi, 2011*) or hormones (primates, *Saltzman & Maestripieri, 2011*)

Corresponding author
Stephen R. Ross, sross@lpzoo.org

that influence maternal care. Studies of social effects, including both maternal and non-maternal influences, have focused primarily on the impact of conspecifics in an infant's environment and the subsequent effect on behavior expressed later in life (humans, *Sroufe, 2005*; rodents & primates, *Pryce et al., 2005*; primates, *Suomi, 1997*). Even animals typically considered less social in nature, such as lizards, seem vulnerable to the effects of atypical and impoverished social histories (*Cissy, Richard & Mats, 2014*).

A range of circumstances in captivity might require infants to be raised by humans rather than by peers or their mothers. For instance, a biological mother may be unable to care for her offspring due to illness or disinterest leading to neglect; in these circumstances, human intervention may be warranted. Studies investigating the impact of these atypical rearing situations on primates—notably with rhesus macaques and chimpanzees—have used categorical classifications, such as mother-reared and human-reared, and reported a range of substantive impacts on developmental trajectories (*Anderson & Mason, 1974*; *Maki, Fritz & England, 1993*; *Suomi, 2011*). In the most extreme cases of social deprivation, there are a range of impacts including behaviors such as self-clutching, excessive rocking, and self-mutilation (*Harlow, Dodsworth & Harlow, 1965*). In laboratory settings, peer-reared chimpanzees (with no maternal contact) tended to be less dominant and less active and often showed more abnormal rocking and less social play behavior (*Rosenblum & Kaufman, 1968*; *Bloomsmith, Lambeth & Alford, 1991*; *Spijkerman et al., 1994*; *Spijkerman, 1996*). Alternatively, a recent study assessing orphaned and mother-reared chimpanzees in African sanctuaries found that orphaned chimpanzees engaged in more social play compared with those classified as mother-reared. However, the play periods of the orphaned chimpanzees were shorter and more often led to aggression (*Van Leeuwen, Chitalu Mulenga & Lisensky Chidester, 2014*). The effects of these early rearing histories may have long-term effects as well: studies of laboratory-housed chimpanzees that were raised exclusively by humans exhibited more abnormal behaviors later in life in comparison to peer-reared chimpanzees who were raised with other chimpanzees (*Martin, 2002*; *Martin, 2005*). In sum, the results of these studies show human-reared chimpanzees seem particularly prone to both short- and long-term negative impacts on their behavior and likely their wellbeing.

Although previous studies have noted both short- and long-term impacts on chimpanzees who were human-reared, past research on human-reared chimpanzees has not taken into account differences in the degree and amount of human interaction in early development and how this influences later behavior. However, social influences may vary depending on an individual's living situation, so a strict categorical classification may not be adequate in many cases. For example, the largely unstudied population of privately-owned chimpanzees—those living as personal pets and as trained performers in the entertainment industry—often includes individuals who have experienced a mix of human and chimpanzee influences. A continuous metric that allows researchers to account for variation in social influences at different stages of an individual's life may be more appropriate than a categorical metric, so that a wide variety of potential influences can be assessed.

Chimpanzees bred for the pet industry are typically removed from their mother soon after birth to facilitate human handling. They are sold to members of the general public, who most often have no experience or training to care for this species. These pet chimpanzees are likely to have relatively little exposure to conspecifics early in life and during key developmental periods. Many are essentially raised as humans, with related traditions such as eating at a table and wearing clothes, until they grow to be too large and dangerous to be kept in the home. Performing chimpanzees have a more variable trajectory and though they may spend some proportion of their time with other chimpanzees, they are also highly exposed to humans (trainers and audiences) until they too typically grow to be unmanageable in adolescence. The categorical designations of "human-raised", "peer-raised", or "mother-raised" do not encompass the actual experience of a chimpanzee such as those who performed and spent significant time in full contact with both humans and other chimpanzees. There is growing consensus that privately-owned chimpanzees represent significant human health and safety risks (*McCann et al., 2007*). In the case of entertainment chimpanzees, these practices produce additional consequences such as negative public perceptions that can impact conservation efforts (*Ross et al., 2008*; *Ross, Lonsdorf & Vreeman, 2011*; *Schroepfer et al., 2011*). However, here we focus on the long-term outcomes of these practices and the degree to which the atypical early histories experienced by these chimpanzees influence their behavioral development.

In the current study, we used a novel approach to assess the impact of atypical early histories experienced by pet and performing chimpanzees and the subsequent outcomes for behavioral development. We employed a long-term continuous measure, chimpanzee-human index (CHI), that accounts for both the amount of time spent with humans as well as the amount of time spent with other conspecifics. We examined how differential human/conspecific exposure during the infant period or the first four years of life impacted current behavioral patterns for ex-pet and ex-performer chimpanzees now living in accredited zoos and sanctuaries. Although our question revolved around chimpanzees without early exposure to conspecifics, we also studied the behavioral patterns of chimpanzees who have lived their entire lives in their natal group to provide a comparison group. These zoo-born chimpanzees may also have a range of human/conspecific exposure early in life; however, most have never had full contact with humans and have always lived with other chimpanzees. In the current study, we focused on the first four years of life because this is considered to be a particularly influential period for behavioral and socio-cognitive development for this species (*Tomasello, Kruger & Ratner, 1993*; *Bard, 1995*; *Bard et al., 2014*). We predicted that chimpanzees with atypical early histories (high human exposure, low conspecific exposure) would differ behaviorally compared with those having more species-typical histories. We projected these differences would be most pronounced in areas of social, sexual and abnormal behaviors and ultimately reflect our hypothesis that these atypical early histories can result in a long-term decrease in species-typical developmental trajectories.

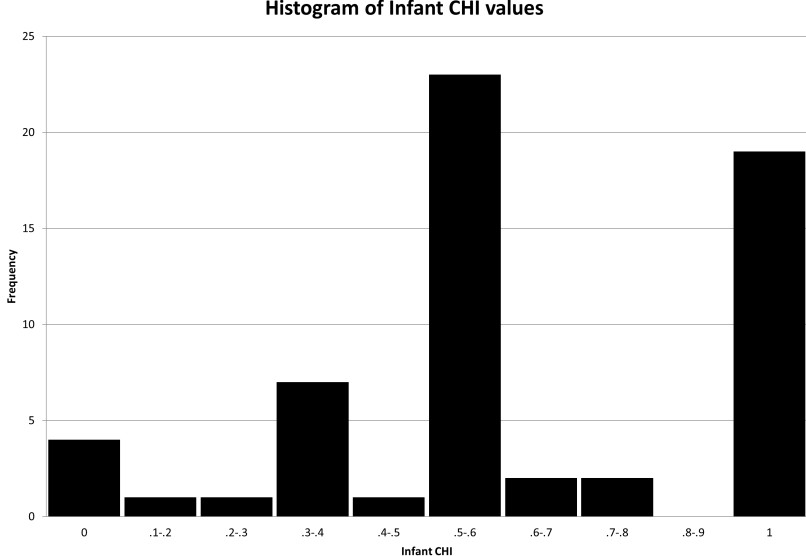

**Figure 1** **Histogram of infant CHI values.** Distribution of $CHI_i$ values across chimpanzees in the study.

## MATERIALS AND METHODS

This research was conducted at three sanctuaries that are members of the North American Primate Sanctuary Alliance (NAPSA: Center for Great Apes, Chimps, Inc., and Save the Chimps) and six zoos accredited by the Association of Zoos and Aquariums (AZA: Houston Zoo, Dallas Zoo, Lincoln Park Zoo, Lion Country Safari, North Carolina Zoo, and Oakland Zoo). Each of these facilities adheres to high standards of chimpanzee care including housing chimpanzees in social groups and providing nesting material and various forms of enrichment. All animals were observed in their home cages during observations. All subjects had *ad libitum* access to water and at no time were the subjects ever food or water deprived. Subjects were provided daily with primate chow and fruit and/or vegetable food enrichment at each of the facilities. This study was approved by and complied with protocols approved by the Chimpanzee Species Survival Plan (SSP) management group as well as animal care committees at each of the institutions that participated in this study.

## SUBJECTS

The subjects were 60 chimpanzees (25 males, 35 females, mean age = 21 years, range: 6–54 years) that varied widely in the degree of human and conspecific exposure they experienced early in their lives (Fig. 1, Table S1). Though many of them were formerly housed as personal pets or performers, all were evaluated in their current housing at NAPSA sanctuaries or AZA zoos. The number of subjects studied at each institution ranged from four to 12. The subjects had been at his/her respective facilities between one month and 54 years before the study. The average amount of time that each subject spent at his/her respective facility before the study was nine and a half years. Data were collected at three NAPSA sanctuaries as well as six AZA-accredited zoos. None of the

NAPSA sanctuaries were open to the public, but the AZA zoos all had public access with daily visitors. All of the subjects were socially housed, with between one and 25 other chimpanzees (average group size was seven chimpanzees). All of the chimpanzees in the study were captive-born, and none had lived in a laboratory environment.

## BEHAVIORAL ASSESSMENTS

Behavioral data on the chimpanzees were collected with a modified version of an ethogram used for Lincoln Park Zoo's long-term behavioral monitoring studies of great apes (*Ross et al., 2011*). The ethogram included six primary behavioral categories (social (e.g., grooming, playing, begging, embrace) sexual, agonism, solitary, inactivity and abnormal), which were comprised of 21 behaviors (provided in the Supplemental Information 1). Behavioral data were collected using a combination of all occurrence and scan sampling. Data were collected with Pocket Observer 2.0 (Noldus Information Technology, Wageningen, The Netherlands) in 30-min focal samples with a 30-s intersample interval for the scan samples. When more than one focal sample was collected on the same individual in one day, there was at least a one hour period between each of the samples collected for that individual. Observations were conducted between 9 am and 5 pm by a single observer (HF) at eight of the nine study sites. The data collected at the ninth site (Lincoln Park Zoo) were collected by observers who had previously achieved 85% reliability on the ethogram. HF conducted a *post hoc* inter-rater reliability assessment with a researcher at the Lincoln Park zoo and was found to have 90% reliability on the ethogram. An equal number of observations were collected in the morning and afternoon for each chimpanzee. The order of the observations was randomly selected ahead of time. Feeding and enrichment times varied at each of the study sites, and data were collected to cover each of these periods. Observations were conducted from a safe and approved area from where the chimpanzees could be easily seen but would not be unduly affected by the presence of the observer. Between 14.5 and 30 h of data were recorded on each subject, over a period ranging from three to eight weeks between November 2011 and November 2012, for a total of 1385 h of behavioral data. The number of hours varied due to time constraints and the number of subjects on which data were collected at each institution. The range in the number of hours collected on each subject was factored into the analyses.

## CHIMPANZEE HUMAN INTERACTION INDEX

In order to characterize the variable degree of exposure to potential influences (conspecifics and humans), we developed a novel, continuous measure, the Chimpanzee Human Interaction (CHI) index. We used management records acquired from past and current holding institutions and calculated the proportion of time per day that each chimpanzee spent in each of three categories: full exposure to conspecifics, full exposure to humans, and mixed exposure to both conspecifics and humans. Each day was assigned a numerical value based on these three categories of exposure. For instance, chimpanzees living exclusively within a large social group in a zoo with only minimal exposure to humans would have a proportion of 1/1 for the day, indicating they spent 100% of their time with other chimpanzees. Likewise, a pet chimpanzee raised exclusively with a

human family, without any exposure to other chimpanzees, would have a proportion of 0/1 for that same day. A performing chimpanzee with relatively equal exposure to small groups of conspecifics and full contact with human trainers and audiences would have a proportion of 0.5/1. CHI is calculated as the sum of these variable exposure periods over a particular timeframe given that many chimpanzees have experienced variation in the degree of human and conspecific exposure across their lifetimes. For this analysis, we chose to focus specifically on the infant period: the first four years of life (see Fig. 1 for a histogram of the infant $CHI_i$ ($CHI_i$ refers to the CHI value of chimpanzees during the infant period) distribution and Table S1 for details about each subject), however the CHI index could be utilized to characterize human/conspecific exposure across any particular timeframe or across the entire lifetime.

## CATEGORICAL GROUPS

Although our CHI index was developed to be used as a continuous variable, $CHI_i$ values in this study revealed a non-normal distribution of the data with clear peaks (Fig. 1). This was likely a result of relatively low variability in conspecific/human exposure over the first four years of life. Therefore, we analyzed the data using categorical groupings by early history experience: with subjects categorized as having only or primarily human exposure ($CHI_i$ index: 0–0.30, $n = 6$), exposure to both chimpanzees and humans ($CHI_i$ index: 0.31–0.70, $n = 32$), or primarily chimpanzee experience ($CHI_i$ index: .71–1.0, $n = 21$) (see Table S1 for each subject's rating).

## DATA ANALYSIS

Data analyses were conducted in IBM SPSS 20 (Armonk, NY). A false discovery rate correction was performed on the results to control for multiple comparisons, and a corrected alpha value of $p$ less than or equal to .01 was considered to be significant for all tests. We controlled for the difference in the number of hours spent observing each chimpanzee by calculating the proportion of time out of the total that an individual spent engaging in a particular behavior compared to the total number of hours of observations on the individual.

In order to assess differences between the three categories of early history during the infant period ("human", "mixed", and "chimpanzee"), we performed a one-way multivariate analysis of variance (MANOVA) using each of the 21 behaviors as dependent variables and the early history categories as a fixed factor.

## RESULTS

One-way MANOVA analysis revealed there was a statistically significant difference in the proportion of time the chimpanzees spent engaging in different behaviors based on their early exposure history ($F(44, 70) = 3.180$, $p < .00005$, Wilks' Lambda $= .111$, partial $\eta^2 = .667$). Separate analyses revealed that sex, age, potential number of social partners, and time spent at current location did not significantly change the results ($p > .05$ for all behaviors after controlling for the effect of early history). A series of one-way ANOVAs on each of the 21 dependent variables was conducted as follow-up tests to the MANOVA.

**Table 1 Significant ANOVA tests with posthoc paired comparisons for behaviors between early history categories during the infant period.**

| Behavior | Early history category | Mean percentage of time engaged in behavior | SD | Early history differences | F-value | Dunnett's T3 Pairwise comparisons |
|---|---|---|---|---|---|---|
| Groom give | Human | 2.40 | 3.27 | H–M | 10.132 | .930 |
| | Mixed | 3.20 | 3.65 | H–C | | .002* |
| | Chimpanzee | 8.61 | 5.73 | M–C | | .012* |
| Groom receive | Human | 1.5 | 1.54 | H–M | 9.219 | .227 |
| | Mixed | 3.1 | 3.20 | H–C | | .000* |
| | Chimpanzee | 6.5 | 3.70 | M–C | | .003* |
| Social sex | Human | .00 | .00 | H–M | 4.983 | .054 |
| | Mixed | .14 | .22 | H–C | | .020* |
| | Chimpanzee | .54 | .76 | M–C | | .084 |
| Inactive | Human | 20.30 | 4.06 | H–M | 7.716 | <.001* |
| | Mixed | 33.22 | 12.20 | H–C | | .886 |
| | Chimpanzee | 22.10 | 10.20 | M–C | | .002* |
| Abnormal coprophagy | Human | .12 | .30 | H–M | 14.994 | .888 |
| | Mixed | .04 | .12 | H–C | | .040* |
| | Chimpanzee | .60 | .58 | M–C | | .001* |

**Notes.**

* indicates a significant pairwise comparison ($p < 0.05$).

An examination of the data revealed that for the following behaviors, the homogeneity of variance assumption was violated: social sex, masturbation, abnormal movement, coprophagy, abnormal body posturing, and abnormal plucking. In these cases, we ran a Welsh test on the data to look at the significance of the ANOVA test. The results of the Welsh test revealed that for the behaviors of abnormal movement and abnormal plucking, the F-value, which was significant in the original ANOVA test, was not significant after correcting for unequal variances. However, the ANOVA tests for the remaining behaviors revealed that in four (social, sexual, inactivity, abnormal) out of six behavioral categories a significant relationship existed between categories of early history and the proportion of time subjects were observed engaging in each behavior. The results of the individual ANOVA tests between proportion of behaviors and categories can be seen in Tables 1–3. In addition, a series of post-hoc analyses (Dunnett's T3 assuming unequal variances) were performed to examine individual mean difference comparisons across the three levels of $CHI_i$ values and each of the behaviors found to be significant with the MANOVA at $p < .05$. The specific results are outlined below.

## SOCIAL BEHAVIOR

The individual ANOVA analyses revealed that early exposure categories influenced frequencies of grooming behavior later in life. The ANOVA revealed a significant difference between groups in frequency of giving grooming ($F(2, 56) = 10.13, p = 0.0001$). Post-hoc analyses revealed that subjects with high amounts of chimpanzee exposure early in life groomed significantly more than those with mixed ($p = 0.012$) or minimal exposure to conspecifics ($p = 0.002$).

**Table 2 Non-significant ANOVA tests for agonism and abnormal behaviors between early history categories during the infant period.**

| Behavior | Early history category | Mean percentage of time engaged in behavior | SD | F-value |
|---|---|---|---|---|
| **Agonism** | | | | |
| Display | Human | .33 | .50 | 1.945 |
| | Mixed | .23 | .23 | |
| | Chimpanzee | .13 | .19 | |
| NC Aggression receive | Human | .00 | .00 | .097 |
| | Mixed | .00 | .00 | |
| | Chimpanzee | .00 | .00 | |
| C Aggression receive | Human | .00 | .00 | .270 |
| | Mixed | .00 | .00 | |
| | Chimpanzee | .00 | .00 | |
| NC Aggression give | Human | .00 | .00 | .064 |
| | Mixed | .00 | .00 | |
| | Chimpanzee | .00 | .00 | |
| C Aggression give | Human | .00 | .00 | 1.157 |
| | Mixed | .00 | .00 | |
| **Abnormal** | | | | |
| Abnormal Movement | Human | 6.08 | 8.60 | 3.492 |
| | Mixed | 1.63 | 4.22 | |
| | Chimpanzee | .79 | 2.50 | |
| Abnormal body | Human | .02 | .03 | .746 |
| | Mixed | .05 | .19 | |
| | Chimpanzee | .00 | .02 | |
| Abnormal Pluck | Human | 2.10 | 3.45 | 6.471 |
| | Mixed | .27 | .45 | |
| | Chimpanzee | .44 | .80 | |

There was also a significant difference between groups in frequency of grooming received ($F(2, 56) = 9.22$, $p = .0001$). Post-hoc analyses revealed that the subjects with high amounts of chimpanzee exposure early in life groomed significantly more than those with mixed ($p < 0.0001$) or minimal exposure to conspecifics ($p = 0.003$). There was not a significant relationship between early exposure category and frequencies of social play or other prosocial behaviors ($p > .01$).

## SEXUAL BEHAVIOR

The ANOVA revealed a significant difference between groups in frequency of sexual behavior ($F(2, 56) = 4.98$, $p = .01$). The post-hoc analyses revealed that chimpanzees with high amounts of early exposure to conspecifics demonstrated higher frequencies of mounting and sexual exploration compared to chimpanzees with low exposure to conspecifics ($p = .02$). There was not a significant difference found between chimpanzees with high exposure to conspecifics and those with mixed exposure ($p > .01$). There was

**Table 3 Non-significant ANOVA results for solitary, social and sexual behaviors between early history categories during the infant period.**

| Behavior | Early history category | Mean percentage of time engaged in behavior | SD | *F*-value |
|---|---|---|---|---|
| **Solitary** | | | | |
| Submissive | Human | .00 | .00 | 3.947 |
| | Mixed | .00 | .00 | |
| | Chimpanzee | .00 | .00 | |
| Attention | Human | 11.92 | 7.12 | 1.144 |
| | Mixed | 12.00 | 5.68 | |
| | Chimpanzee | 9.09 | 7.12 | |
| | Chimpanzee | .00 | .00 | |
| Self play | Human | .11 | .11 | .579 |
| | Mixed | .29 | .71 | |
| | Chimpanzee | .14 | .40 | |
| Self groom | Human | 17.46 | .07 | 2.821 |
| | Mixed | 13.87 | .05 | |
| | Chimpanzee | 12.40 | .03 | |
| Locomotion | Human | .03 | .01 | 1.655 |
| | Mixed | .03 | .01 | |
| | Chimpanzee | .03 | .01 | |
| **Social** | | | | |
| Social play | Human | .42 | .46 | 2.088 |
| | Mixed | .98 | 1.26 | |
| | Chimpanzee | 1.54 | 1.48 | |
| Prosocial | Human | .14 | .15 | 3.640 |
| | Mixed | .16 | .13 | |
| | Chimpanzee | .37 | .44 | |
| **Sexual** | | | | |
| Sex masturbate | Human | .00 | .00 | 3.946 |
| | Mixed | .11 | .00 | |
| | Chimpanzee | .00 | .00 | |

not a significant relationship found between the category of early exposure and rates of masturbation ($p < 0.01$).

## AGONISTIC AND SOLITARY BEHAVIORS

There was no difference in the expression of agonistic or solitary behaviors, including displays, non-contact, contact aggression given or received, self-grooming or self-play in relation to categories of early exposure ($p > .01$).

## INACTIVITY

The ANOVA analyses revealed significant differences in frequencies of inactivity between the early history categories ($F(2, 56) = 7.72, p = .001$). Post-hoc tests revealed that chimpanzees with mixed early exposure demonstrated higher rates of inactivity than

those with either primarily human ($p < .0001$) or primarily conspecific exposure early in life ($p = .002$).

## ABNORMAL BEHAVIOR

We examined four forms of abnormal behavior: coprophagy, abnormal movement, abnormal body posturing, and hair plucking. The ANOVA analysis revealed significant differences in frequencies of coprophagy between the early history categories ($F(2, 56) = 14.99, p = .0001$). The post-hoc tests revealed that chimpanzees with high amounts of early exposure to conspecifics demonstrated the highest frequencies of coprophagy later in life than those with either mixed early exposure ($p = .040$) or primarily human exposure early in life ($p = .001$). There was not a significant difference in rates of coprophagy between chimpanzees from primarily human or mixed early exposure histories. The ANOVA analysis found there was not a significant difference between the three categories in rates of the other forms of abnormal behavior ($p > .01$).

## DISCUSSION

The primary aim of this study was to use a novel approach to assess the long-term behavioral impact of variable early life exposure to both conspecifics and humans on captive chimpanzees. To achieve this, we focused on a rarely studied population, former pet and performer chimpanzees that now live in accredited zoos and sanctuaries with other chimpanzees. We also studied individuals who had experienced more typical early life histories for captive chimpanzees, living in their natal group with multiple conspecifics, in accredited zoo environments throughout their lifetime. In general, supportive of our hypotheses, we found that chimpanzees raised in "human" or "mixed" exposure groups tended to exhibit lower frequencies of grooming compared with those in the "chimpanzee" group. In addition, chimpanzees raised in the "human" exposure group exhibited lower frequencies of social sexual behavior later in life as compared with those in the "chimpanzee" group. Lastly, chimpanzees in the "mixed" exposure group exhibited higher frequencies of inactivity compared with the "human" or "chimpanzee" groups. The results of this study suggest that high human exposure early in a chimpanzee's life, and/or reduced conspecific exposure, is related to observable differences in behavior in adulthood. Grooming and sexual behavior are important components to the dynamics of social groups in chimpanzees. Decreases in these behaviors could have the potential to be related to animal management and welfare issues connected with social interactions between chimpanzees.

While it is possible that variability in current living environments may represent a confound to the conclusions of this study (due in part to the relatively limited number of institutions caring for chimpanzees with such atypical backgrounds), it is important to note that the demonstrated effects are unlikely to be solely the result of current physical or social environments. All subjects were socially housed, and all were managed under similar contemporary management systems.

The behavioral differences observed between early-history groups are consistent with previous findings in chimpanzees and other non-human primates (i.e., mother-reared

individuals compared to isolated individuals, *Turner, Davenport & Rogers, 1969*; *Davenport & Rogers, 1970*; *Harlow & Suomi, 1971*; *Kalcher et al., 2008*). For example, several studies have demonstrated that human-reared primates often have lower rates of reproductive success, possibly indicative of less appropriate sexual behavior (*Ryan, Thompson & Roth, 2002*). The evidence for a developmental influence of rearing on later chimpanzee behavior is mixed. While some studies report a lack of rearing effect in social behavior (*Bloomsmith et al., 2002*; *Howell et al., 2006*; *Van Ijzendoorn et al., 2009*) more recent analyses (*Clay, 2012*) and the results of our analyses suggest the possibility that these effects might not be evident until much later in life. Duration of exposures of chimpanzees to humans or conspecifics might also be a particularly important factor. *Martin (2005)* assessed behavior in chimpanzees that were reared in a variety of contexts and found no statistical effect of rearing; however, the authors were unable to account for the duration of time spent in each of the rearing categories.

We also noted differential effects of early experiences on coprophagy, a behavior typically categorized as abnormal (*Walsh, Bramblett & Alford, 1982*; *Nash et al., 1999*). We found that chimpanzees with more conspecific exposure engaged in coprophagy more frequently compared to those in the "mixed" or "human" groups. This behavior is likely socially learned (as opposed to an individually-developed response indicative of stress) and thus animals with broad social exposure may be more likely to adopt these behaviors. Previous studies have found that mother-reared chimpanzees engage in coprophagy more often than non-mother-reared individuals (*Nash et al., 1999*; *Bloomsmith et al., 2005*), which suggests that coprophagy may be functionally distinct from other abnormal behaviors. Future studies should involve investigating potential welfare concerns associated with different types of abnormal behaviors in order to determine which behaviors are performed only during times of stress compared with other times. Despite the breadth of the data used for this investigation, there remain a number of limitations that reflect the complexity of a multi-institutional study and require additional consideration. The first is the potential confound between a chimpanzee's current housing locations with his/her early history exposure. Chimpanzees from particular backgrounds tended to cluster at specific institutions. As a result, there was relatively little intra-institutional variation in early histories among the chimpanzees housed within single groups. Fortunately, the variation in current housing conditions and management styles was relatively low, at least in comparison to the wide variety of physical and social environments from which many of these chimpanzees originated. All current housing was either at AZA-accredited zoos or NAPSA member sanctuaries; all maintain consistently high standards of care in terms of diet, enrichment, and housing. Additionally, subjects were housed in different social groupings (ranging from 2–26 chimpanzees per group), and in some cases, this was an outcome of their atypical early histories and difficult socialization histories. In addition, some of the chimpanzees were genetically related to each other; however, because of the amount of variability in the degree of relatedness both within and across institutions, we could not account for it given our current sample size.

We must also consider the potential limitations of our measure of human and conspecific exposure, including both the CHI index and categories based on this index. In this analysis we utilized a subset of the CHI index ($CHI_i$), specific to a relatively narrow time frame from birth to age 4. During this timeframe, there is inherently less variation across individuals in comparison to that observed across the entire lifetime. For instance, chimpanzees with low conspecific exposure were usually maintained in those conditions across those first four years, but across their lifetime they may experience a wider array of social conditions. Subsequently, we chose to use a categorical analysis. Only six of the 60 chimpanzees had solely human exposure during infanthood but including this category in the analyses was important to draw comparisons among those individuals who lacked exposure to conspecifics. Inclusion of this small sample group comes with limitations including the possibility of failing to detect some behavioral effects (type II errors). We also acknowledge that it is possible that individuals with similar CHI values may in fact have experienced very different patterns of exposure to conspecifics and humans. Further refinement of this index may help address these potential limitations.

Overall, the results of this study suggest that adolescent and adult chimpanzee behavior is associated with early life experiences and that individuals exposed to more human-centric environments may express behavioral deficits later in life in relation to grooming and sexual behavior. It is possible that the consequences of these developmental trajectories may differ considerably for chimpanzees living in situations with greater regular exposure to humans, such as in laboratory settings. Research on how the CHI index relates to differences in personality, cognitive performance, stress levels (as measured through cortisol), and health histories is likely to give a more comprehensive representation of how these effects are manifested. Finally, care should be taken to consider the substantive variation in how these early histories affected individuals. Not every chimpanzee who had a low $CHI_i$ score showed deficits in social and sexual behaviors, as evidenced by the wide-ranging standard deviations. Future studies should help identify what variables lead to better social resilience in order to aid chimpanzees who struggle more with social integration. The results of this research suggest that future research should focus on developing the best management strategies for how to care for chimpanzees with a variety of early histories in order to meet their social needs. The reduced (or absent) exposure to conspecifics and full contact exposure to humans that these chimpanzees experienced, especially during the first four years, may have especially profound and long-term behavioral outcomes. Given the known public safety concerns surrounding pet chimpanzee ownership and the negative perception and conservation impacts of inappropriate media portrayals of privately-owned "actor" chimpanzees (*McCann et al., 2007*; *Ross et al., 2008*; *Ross, Lonsdorf & Vreeman, 2011*), we now add empirical evidence of the potentially negative welfare effects on the chimpanzees themselves as important considerations in the discussion of privately-owned chimpanzees. We promote further use of these and other evidence-based methods to further inform policy and legislative change that protects chimpanzees and other important non-human animals that are subject to conservation and welfare threats.

## ACKNOWLEDGEMENTS

We are grateful to Lydia Hopper and Joe Simonis for their help in refining the Chimpanzee/Human Interaction index. The manuscript was greatly improved thanks to the thoughtful feedback provided by Lydia Hopper, the journal editors and four reviewers. We are grateful to Kathy Wagner for her help with using the Observer software for behavioral data collection and to Andrew Steets for his computer programming skills. Thank you to the interns at the Lincoln Park Zoo who helped to collect the behavioral data at that facility. We would also like to thank the staff at the Houston Zoo, Dallas Zoo, Lion Country Safari, Center for Great Apes, Save the Chimps, North Carolina Zoo, Chimps Inc. Lincoln Park Zoo, and Oakland Zoo for their support and allowing us to collect data at their facilities.

### Funding

This project was funded by a grant from the Arcus Foundation (grant number 1102-34). The funders had no role in study design, data collection and analysis, decision to publish, or preparation of the manuscript.

### Grant Disclosures

The following grant information was disclosed by the authors:
Arcus Foundation: 1102-34.

### Competing Interests

The authors of this manuscript have no competing interests with submitting this manuscript to PeerJ. This includes not having any financial, non-financial, professional or personal relationships including serving as an Academic Editor on the PeerJ Board. Hani D. Freeman and Stephen R. Ross were employed by the Lester E. Fisher Center for the Study and Conservation of Apes, Lincoln Park Zoo at the time the research was conducted.

### Author Contributions

- Hani D. Freeman conceived and designed the experiments, performed the experiments, analyzed the data, contributed reagents/materials/analysis tools, wrote the paper, prepared figures and/or tables, reviewed drafts of the paper.
- Stephen R. Ross conceived and designed the experiments, contributed reagents/materials/analysis tools, wrote the paper, reviewed drafts of the paper.

### Animal Ethics

The following information was supplied relating to ethical approvals (i.e., approving body and any reference numbers):

This study complied with protocols approved by the Chimpanzee Species Survival Plan (SSP) management group as well as approval at each of the institutions that participated in this study (Lincoln Park Zoo, North Carolina Zoo, Oakland Zoo, Dallas Zoo, Houston Zoo, Lion Country Safari, Center for Great Apes, Save the Chimps and Chimps Inc). Each of these facilities adheres to high standards of chimpanzee care

including housing chimpanzees in social groups, providing nesting material and various forms of enrichment. All animals were observed in their home cages and were not confined to a specific area of their enclosures during observations. All subjects had ad libitum access to water and at no time were the subjects ever food or water deprived. Subjects were supplemented daily with primate chow and fruit and/or vegetable food enrichment at each of the facilities.

## Supplemental Information

Supplemental information for this article can be found online at http://dx.doi.org/10.7717/peerj.579#supplemental-information.

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
