# Peer review of "The impact of atypical early histories on pet or performer chimpanzees"

_PeerJ, doi:10.7717/peerj.579_

## Round 0.1 · original submission · Major Revisions

· Academic Editor

Major Revisions

The reviewers viewed the manuscript favorably but indicated additional work would be necessary before this manuscript is suitable for publication. The reviewer feedback from all three reviewers should guide you in revisions of your manuscript, but let me summarize the key issues noted for each area of PeerJ publishing criteria:

(1) Basic reporting
(a) Some terms require definition or clarification.
(b) The writing style was too informal and sometimes confusing.
(c) Be sure to check grammar (spelling, subject/verb agreement, punctuation, and syntax) before submitting the revised manuscript.
(d) Please make sure the abstract is reflective of the body of the manuscript.

(2) Experimental Design
(a) Additional information on the methods used for making the observations is necessary.
(b) One reviewer expressed concerns about the robusticity of CHI when the period of "early rearing" is defined other than 0-4 years. Please address to what extent other periods alter the value of CHI or explain why such a concern is a non-issue.
(c) Another reviewer expressed concern that CHI does not adequately account for the importance of the mother chimpanzee in chimpanzee rearing and did not adequately account for variation among the sites studied (e.g., level of human interaction in later life, site-specific environmental conditions) and other variables (e.g. genetic variation).

(3) Validity of Findings
(a) The reviewers indicated additional details are necessary to assess the validity of the statistical results. Please include additional statistical information in the manuscript or as supplemental material.

·

Basic reporting

Below are minor comments but they need to be corrected before resubmission.

L129-130. I found confusion in the description of numerals. Authors need to revise them throughout the manuscript by following the style consideration of PeerJ guidline.

L132. The observation period that authors collected behavioral data is not addressed in the method.

L189. “MANOVA calculation” can be integrated into Data Analysis paragraph.

L194- Result section
“MANOVA Analysis” seemingly does not make sense for the subhead. I think it should be changed to more proper subhead. Additionally, there are several short paragraphs such as agonistic behavior, solitary behavior, and inactivity. It is better to revise the result section by following Standard Sections of PeerJ guideline.

L220. Sexual difference in sexual behavior will be informative for readers, if it was detected.

Experimental design

My major concern is the data analysis employed by authors as below;

1) Period of early rearing condition
Authors used the rearing data during 0-4 years old for evaluating early rearing condition. The period of 0-4 years old is corresponding to “infant” of developmental stage in chimpanzees, a representation of early rearing condition. How we can determine the period of EARLY rearing condition in chimpanzees is still unclear, although the previous studies had pointed out the influence of rearing condition during infancy on behaviors of captive chimpanzees. So, there is a question that the results of the study may possibly be different depending on the cases that authors take the data during the period of 0-3 or 0-5 years old as early rearing condition.
It is because CHI is a continuous variable. The value of CHI will vary according to the range of period designated as early rearing condition by authors. Authors can also analyze the relationship of behavioral data with various duration of rearing condition such as 0, 0-1, 0-2, 0-3, 0-4, and 0-5 years old, for example.
Thus, authors need to address the soundness of the result in the study. I would also like to encourage further, in-depth analysis in order to clarify whether there is a crucial period at the beginning of development that chimpanzee’s behavior is affected in later.

2) Categorical analysis of rearing history as human, mix, and chimpanzees
Authors transferred the quantitative data of CHI into a categorical data such as human, mix, and chimpanzee. I would like to point out that the data exchange can be inevitability arbitrary, to some extent. Although authors showed the distribution of the CHI data in Figure1, the distribution will vary according to the period defined as early rearing condition by authors as pointed out above.
In general, the difference between mother-reared and human-reared is not qualitative (discrete), but quantitative (continuous). But, few study analyzed the effect of early rearing condition on behavior using a continuous variable of rearing history. CHI introduced here is a very unique and useful index for quantifying the gradient of mother- and human-reared proportion within a certain individuals.
Thus, I would like to encourage a re-analysis using the continuous CHI variable, rather than using the categorical variable.

Validity of the findings

See the comments at Experimental Design.

Comments for the author

This paper aims to understand on the long-term effects of early rearing conditions on later behavior of captive chimpanzees. The topic matter and aim of the study is important and deserving of investigation for developing the self-sustaining population management of captive chimpanzees. The idea of the study (i.e., CHI) is very unique and interesting which would be a valuable contribution to primatology.
I think this manuscript is ultimately publishable after major revisions.

·

Basic reporting

This article is well organized, well written and highly comprehensible.

I just ask the authors to define a vague term that they repeatedly use in the introduction: "appropriate socialization" (lines 32, 37, 46)

And please indicate in line 137 that the 21 behaviors are defined in the Online Supplementary Materials.

Experimental design

The study is well designed and very comprehensive. The authors amassed an impressive amount of data on a large sample of chimpanzees to answer their questions and with the exception of the chimpanzees at one institution all data was collected by a single individual, reducing potential variation that could owe to differences between observers.

Validity of the findings

The authors have interesting and important findings. However, they have not reported sufficient information for a reader to completely evaluate their statistical analyses or results.

It is essential that the authors include the following information before the manuscript will be ready for publication:
1. (line 186) What false discovery rate correction was used? After correction of false discovery rate, did you only accept as significant a p-value of 0.01? Is this correction specific only to the the MANOVA and ANOVAs?
2. (line 202) How did the Welsh test impact interpretation of the results?
3. (line 208) Please clarify the post-hoc analyses that you used. I'm assuming from your table that these were Tukey's HSD Pairwise comparisons, but this is not indicated anywhere else in the manuscript.
4. (line 208) Please also clarify that you mean that you adjusted the significant p-value for your post-hoc analyses using a Bonferonni correction. What was the correct p-value?
5. Throughout the results section, please clarify the statistical tests for which you are reporting results. I believe that all results reported in the results section are those of individual ANOVAs; however, this is made unclear because these results are reported following descriptions of the results of the post-hoc analyses (e.g., lines 214-216). The p-values of the post-hoc analyses then are only reported in the table and not in the text. Please include this information in the text as well. For example: The ANOVA revealed a significant difference between groups in frequency of giving grooming (F(2,56) = 10.13, p = 0.0001). Post-hoc analyses revealed that subjects with high amounts of chimpanzee exposure early in life groomed significantly more than those with mixed (p = 0.012) or minimal exposure to conspecifics (p = 0.002).
6. Please either separate the table into two tables or add a clear vertical line separating the columns entitled "SD" and "Early History Differences". Your means and SDs are associated with the Early History Category but your Tukey's HSD pairwise comparison p-values are associated with the Early History Differences, and in its current form this is unclear.
7. Please also include more information in the table title and description to help readers understand which columns are associated with which populations (i.e, early history category versus early history differences) and an explanation that H means Human-exposed, M Mixed and C Chimpanzee-Exposed Histories. Also make clear that the F-values are comparing all 3 groups and are not associated with Early History Categories or Early History Differences.
8. Please include in the table the p-values for the results of the ANOVAs.
9. Why are there results of post-hoc testing for ANOVAs that were not significant? If an ANOVA is not significant, post-hoc testing should not be reported.
10. Is the Tukey's p-value for Social Sex M-C a typo? It says that it is significant at 0.084. Similarly, why are p-values of 0.014 and 0.011 not significant for Sex Masturbate H-M and M-C respectively?

Comments for the author

This is a great and incredibly important paper. I am strongly in favor of this paper being published in PeerJ, once the authors have made the suggested edits to more clearly report their statistical analyses and results.

Reviewer 3 ·

Basic reporting

See review

Experimental design

The research was ethical. Additional variables need to be accounted for in the statistical analyses.

Validity of the findings

See below

Comments for the author

The impact of atypical early histories on pet or performer chimpanzees

As pointed out by the authors, the early history of primates can drastically affect their later behavior. Typically, studies of early history focus on subjects reared with their mother compared to those reared with peers or in a nursery. Few studies have factored in the effect of humans. The authors developed a “Chimpanzee-Human Interaction” index as a way to examine early rearing in chimpanzees, and compared CHI with later behavioral outcomes. I think this is a very interesting idea, and particularly liked the fact that it was a continuous measure, since, as the authors point out, too often animals are categorized, somewhat arbitrarily. This methodology is novel and could provide great insight. However, I do have a few concerns about this particular paper.

My first concern is with the CHI. There are a number of studies on rhesus macaques demonstrating that not all conspecifics are created equal when it comes to rearing; animals reared with their mothers tend to show fewer behavioral problems later in life compared to those reared with conspecific peers, for example. In the present study, it appears as though there is no difference between the mother and other chimpanzees in the “C” part of the CHI. A chimpanzee living exclusively in a large social group with her mother would get the same CHI score as an orphaned chimpanzee living in a large group or a chimpanzee who was reared with a peer. The authors might want to explain this index a bit more, particularly given that they spend a great deal of the introduction discussing the importance of the mother in development.

My second concern is with the statistics. There is are many variables that could affect behavior of the subjects, including age, gender, amount of time spent at the current facility, type of current facility (zoo vs sanctuary), enclosure size, number of animals in the enclosure, etc. The authors’ assertion that because all of the sites are accredited they are essentially the same is not really valid. The animals living at a zoo will have a different experience than those living in a sanctuary, if for no other reason than the interaction with people other than their caregivers. If proximity to humans is important in early development (the “H” in CHI), shouldn’t it be assumed to be important later in life? Some of the animals lived in groups of 2 and others in groups of 26; presumably the enclosure sizes varied quite a bit as well. Further, since only one person took the majority of data, data collection presumably occurred at different times of the year. All of these factors should be accounted for in the analyses.

Other, specific comments:
In many places, the writing style seems very casual, with lots of dashes in the middle of sentences, etc. There are also several places in which I found the text a bit hard to follow.

Abstract
In the first sentence, the authors talk about “an animal” (singular) and “their” rearing (plural). These terms should be consistent.

Second sentence: the authors talk about categorical measures, but don’t describe what those are.

Throughout the MS, the authors talk about their approach as “holistic”, but I am not sure that is accurate. Their approach, while novel and interesting, does not take many factors into account, including the presence of the mother, older sisters, or other social support, rank, various environmental factors, etc. The authors should make a stronger case for calling their approach “holistic”.

The third sentence is somewhat awkward.

The abstract specifically mentions the creation of the CHI index, but the only result given is that “chimpanzees who experienced less exposure to other chimpanzees showed lower social and sexual behaviors later in life”. Since the authors set up the abstract as assessing the CHI index, they should consider mentioning it when discussing their results. They had many results other than that one.

The abstract mentions that the results of this study can help inform managers seeking to integrate these types of chimpanzees into groups, but this is not discussed elsewhere in the MS. The abstract should not bring up ideas not presented elsewhere in the paper.

Introduction
In general, I found the introduction to be a bit long and not really focused on the CHI index. The authors spent a lot of time discussing the importance of maternal rearing, but did not specifically mention the CHI index at all. The authors should try to be a bit more focused. Why did they create this CHI index? What does it offer that is different from other methods of examining the effects of early experience on later behavior? They touch on this in the abstract, but should describe it in the introduction.

Line 6-7: This first sentence either contains a typo or is missing a few words, or should read “xxx later in life (). Outcomes xxxx”.

Line 7: the authors mention “the expression of those behaviors later in life”. To which behaviors are they referring?

Line 32: What do the authors mean by “unplanned circumstances”?

Line 36-37: What do the authors mean by “the most extreme lack of appropriate early socialization”. The authors mean that the infants were reared in social isolation. They should try to be more direct. Phrases such as “the most extreme lack of appropriate early socialization” are somewhat awkward.

Line 44: What do the authors mean by “less severe environments”? There are many studies in rhesus that are also “less extreme” than early Harlow work (e.g., those by Melinda Novak, Ruppenthal, Sacket, Suomi, just to name a few).

Line 57: The phrase “with conspecific involvement in rearing” is awkward.

Line 59: What do the authors mean by “typical early histories”? Typical for animals in a captive environment or typical for wild animals?

Line 60: The authors summarize their paragraph by stating that animals reared without maternal and conspecific support are “prone to both short and long term negative impacts on their behavior”, but they haven’t specifically stated that in the preceding paragraph. The authors have stated that chimpanzees reared in this type of environment played less and showed more rocking, but that is not really the same as saying that they are prone to short and long term negative impacts of behavior. There are many studies in rhesus that show these kinds of long term outcomes, but they are not mentioned in this paragraph. I actually think much of this paragraph can be reduced, as I think most readers understand the importance of the mother in early development.

Line 76: I don’t know if the authors really know whether members of the general public lack formal husbandry training. Most don’t, but it is possible that some do. The authors should avoid making strong statements that they cannot support.

Line 77: What do the authors mean by “key developmental stages”? They haven’t yet brought up the idea of “key developmental stages”.

Line 78: What do the authors mean by “human traditions”?

Line 87: What do the authors mean by “ontogenetic outcomes”? Since the subjects are a variety of ages, and have presumably been out of their previous environment for varying amounts of time, it is hard to study specific developmental outcomes.

Line 96-98: I’m not sure what the authors mean by “We acknowledge that the impact of these different early histories could vary for chimpanzees entering various environments xxx”.

Line 99-101: This sentence doesn’t make sense to me. Are the authors basically stating that they also studied animals with a more normal upbringing as controls?


Methods
Much more detail is needed about the specific sites as well as individual histories of the animals. How big were the enclosures? How much, if any, exposure to human visitors did they have? How long had they been at their respective institution prior to observations? An animal moved around a lot might behave differently than one born and raised at one institution. Were any of the subjects related (e.g., mom and offspring)? Since many behaviors have a genetic component, that could affect the results.

What kind of sampling methodology was used to take observations? Behaviors with relatively short durations, such as aggression, tend to be missed with instantaneous sampling with relatively long inter-sample intervals (e.g., 30 sec). Why was there such a high degree of variation in the amount of observations taken per individual? Was this accounted for in the analyses?

Line 157: Did chimpanzees from private homes or entertainment facilities have management records?

Line 187: The alpha value is typically adjusted; the p value isn’t typically corrected. This sentence is somewhat awkward.

Line 192: in this paragraph, the authors state that there are 22 behaviors, but in line 137, they state 21. There are 19 behaviors listed on the ethogram. These should be consistent.

Line 193: Perhaps I don’t understand MANOVAs as well as I should, but I don’t understand how behavioral categories, which are somewhat arbitrary are a fixed factor in the analysis.

Results
As mentioned above, without having accounted for sex, age, current location, etc, it is hard to really evaluate the findings.

Discussion
The authors do a nice job discussing the limitations of their study.

Line 263: Since the authors only studied the chimpanzees at one time point, they can’t really say that early events resulted in differences in behaviors that were “sustained throughout their life”. They may have observed the animal 20 years after the adverse rearing. Perhaps some individuals showed a great deal more abnormal behavior (for example) when they were younger. As an example, the authors referenced a paper that found that orphaned chimpanzees were more likely to have play behavior end in aggression than non-orphaned chimpanzees. But, in the current paper, they found no difference in aggression. Perhaps that is because differences in aggression were not sustained throughout their lives. The authors should avoid making statements for which they have no data.

Line 272: What do the authors mean by “different groups”?

Line 294-295: What do the authors mean by “during times of stress compared with times of boredom or during routine times”? What is a routine time?

---

## Round 0.2 · Minor Revisions

· Academic Editor

Minor Revisions

The three original reviewers agree that your revisions substantially improved the manuscript. Because the reviewers did not have specific expertise in statistics and in light of the questions raised by one of the original reviewers about the appropriateness of the methods in the revised submission, it was necessary to invite a fourth reviewer with statistical expertise to ensure that the methods used were appropriate and the results valid.

The major problems noted in first round of reviews have been resolved, but there are a few remaining minor issues regarding the basic reporting that need to be addressed before this manuscript is ready for publication. Please note that any line references included in my comments correspond to the reviewing .pdf (not the .doc) version.

(1) Clarification of terms needed.
There are a few points in the text where one or more reviewers noted your meaning of phrases are unclear, such as the meaning of “particularly influential period” (line 99), “detrimental developmental trajectories” (line 104), and “developmental influence on rearing” (line 288). Please clarify where appropriate.

(2) There are numerous grammatical problems throughout the manuscript. Please review your manuscript carefully from beginning to end to correct them. Examples of the grammatical problems include the following:
- Antecedent/Object disagreement and apostrophe misplacement (e.g., line 21 an offsprings’)
- Run-on sentences (e.g., lines 30-32; lines 148-149; 171-175)
- Unclear antecedents (e.g., lines 51-55 “atypical” is introduced as shorthand for what? The parenthetical note and extremely long sentence is confusing)
- Use of numbers in text (e.g., line 125 “twelve” should be “12.” Do not spell out numbers ten or greater unless the number is starting a new sentence; line 169 would be better as “0.5/1” rather than “.5/1”).
- Misuse of commas (e.g., line 150 where the comma is extraneous; lines 166-167 where the clause “without any exposure to other chimpanzees” should be flanked by commas rather than simply preceded by one comma; line 198 where a comma is missing after the third item in the four-item list; line 202-204 where a comma is missing after the fifth item in the six-item list)
- Subject/pronoun disagreement (e.g., line 126 “each subject…their facility” and again in lines 162-165 “a chimpanzee…they spent” where a plural pronoun was used to describe a singular subject.)

I look forward to receiving your minor revisions.

·

Basic reporting

I agree that the description is much improved throughout the manuscript.

Experimental design

I agree that the description is much improved.

Validity of the findings

I agree that the description is much improved.

Comments for the author

Ref.: Ms. No. Peerj-1728


I agree that the paper is much improved and nearly ready to be accepted. I would recommend accepting the paper by Drs. Freeman and Ross, “The Impact of Atypical Early Histories on Pet or Performer Chimpanzees”, for publication in The Peerj, after the minor revision.

L129 I think "12" instead of "twelve".

L216 I think “inactivity” instead of “active”. I also think each of the six behavioral categories can be deleted, because those were written in the method section.

L248-261 I would like to point out the redundant description in the Result section. The sub-sections of Agonistic Behavior, Solitaly Behavior, and Inactivity can be combined for a simple description.


Best wishes,
Naruki Morimura, Ph.D.

·

Basic reporting

The authors have adequately addressed all of my concerns.

Experimental design

The authors have adequately addressed all of my concerns.

Validity of the findings

The authors have adequately addressed all of my concerns.

Comments for the author

The authors have adequately addressed all of my concerns. As noted in my first review, I believed that this article makes an important contribution. With the changes made to the manuscript to clarify statistical methods and the reporting of results, I advocate for its publication.

Reviewer 3 ·

Basic reporting

See below

Experimental design

No comments

Validity of the findings

See below

Comments for the author

The authors have, for the most part, done a nice job with their revisions. However, there are still many places in which the text is somewhat awkward and hard to follow. There are also many errors with punctuation. The authors might want to have someone go through this MS carefully. Other, more specific comments are detailed below.

Introduction
While the authors have tightened up this section, and have done a nice job introducing the idea of CHI, in my opinion they still spend a bit too much timediscussing the importance of maternal attention, given that it is not a variable they examine. Much of the first, the entire second and third and much of the fourth paragraphs focus on the role of the mother in the development of NHPs. This discussion is somewhat distracting, because it really does not address the main point of this paper, which is the effect of conspecific and/or human interaction. As I mentioned in my last review, the emphasis placed on the importance of being reared with the mother leads one to wonder why that was not included as a variable in this study.

Line 10-11: I am not sure what is different between “behavioral development” and “expression of behavior later in life”.

Line 25: The authors should change “offsprings’” to “offspring’s”.

Line 31-32: What do the authors mean by “a range of circumstances, planned and unplanned…”. Who is planning or not planning these circumstances? The authors give examples, but this is still an awkward way of saying that infants might be reared without their mothers for naturally occurring and experimental purposes.

Line 57- What is hereafter termed atypical- the type of rearing?

Line 58: Previous social experiences may certainly influence future success in group living. The authors bring this idea up, but do not provide any kind of evidence to suggest that there was a difference in ability to live in a group between animals with low and high CHIs. The only social behavior data they seem to have examined was grooming, aggression, etc. The authors could describe how the data they collect may influence an individual’s ability to live in groups, but as it stands, this seems out of place because it is not examined.

Line 62- Ultimately, the authors also categorized their subjects by rearing group, so they may not want to be so hard on other studies that used categories. What is different about their study is really the inclusion of interaction with humans, not the lack of categorical data.

Line 98: I am not sure what the authors mean by “our question revolves around chimpanzees without early exposure to conspecifics”. In the previous sentence they mention that they examined the effects of differential human/conspecific exposure. In fact, at no point do they mention that their question revolved around human reared infants.

Line 100 should be past tense.

Line 103: What do the authors mean by “particularly influential period”? Influential in what way?

Line 107-108: What do the authors mean by ‘detrimental developmental trajectories”? Further, the authors do not really talk about their results in terms of well being. If altered well being is their hypothesis, they should describe how they are measuring wellbeing in their subjects.

Methods

In the response to reviewers, the authors state that they clarified methodology for behavioral observations that “some of the behaviors such as aggression were recorded as all occurrences and others as scan”, but I could not find this in the revised MS. Further, while the authors stated in their response letter why there were large discrepancies in the amount of observations and data across centers, they do not mention this in the MS. They should, as it currently seems rather arbitrary as currently written. The authors explain briefly in the methodology that these differences were accounted for, but they should describe how these differences were actually accounted for in the data analysis section.

Line 110-114- This sentence is quite long and hard to follow. The authors might break it to two sentences, one that states that research was performed at 3 sanctuaries and 6 zoos (with the names) and the other that states something to the effect of “all sanctuaries were members of NAPSA and all zoos were AZA accredited.

Line 116-117: Confining subjects to specific areas of their enclosures is not common practice for behavioral observations, so it is not clear why this was brought up.

Line 118: Supplements are typically given in addition to the normal food; was primate chow the main food source or was it given in addition to their main food? If the latter, then the main food source should be listed. If primate chow was the primary food source, then it may have been provisioned, but was not supplemented.

Line 122: Funding information should go in acknowledgements

Lin e130: “each subject” is singular, but “their” is plural. “Their” should be changed to “its” or “his/her”.

Line 134-135 There is no reason to state “who participated in the study”, since you are talking about subjects. I would remove “who participated in the study” from this sentence.

Line 150: Was the researcher with whom HF was reliable one of the observers who previously took data on the chimpanzees? In the acknowledgements, it seems as though interns, not researchers, took the observations.

Line 165: I’m not really sure what is meant by “each day was weighted”; the animals got a score of 0, .5, or 1 for each day. That isn’t really being weighted.

Line 173-175: This sentence is awkward and hard to follow.

Line 177: This is the first time the authors use the subscript “I” after CHI, so they should explain what that is.

Line 188- Is the “primarily chimpanzee experience” group the control group?

Line 190-191: This sentence should be in the discussion.

Data analysis- The authors need to include more information in this section. They mention statistical tests in the results that they do not mention here (e.g., Welsh test). Also, this is where they should discuss how they controlled for the varying amount of observation time across facilities. They also state in their response to the reviewers that they examined factors such as gender, age, current location, etc.; this information should be provided here as well.

I am also not sure why they needed to adjust their alpha if they were doing a MANOVA. Doesn’t the MANOVA take multiple observations into account? MANOVA also takes into account correlations between the dependent variables, yet this is not mentioned in the results.

Results
Line 206: should the <.05 be greater than 0.05?

Why did the authors do a series of univariate ANOVAs if they had done a MANOVA? In SPSS, isn’t there a step down series of ANOVAs that are really more like post-hoc analyses?

Looking for assumptions of normality, and homogeneity of variance should be done before analysis, not after. If there were variables that did not meet these assumptions, they should not have gone into the overall MANOVA.

I am not really clear on how the authors did their analysis. Why did they use a MANOVA if they did not look for relationships between the DVs? Further, I don’t know how one does a post-hoc test (line 220) on a post hoc test (a univariate ANOVA is similar to a post hoc analysis for a MANOVA).

I am not familiar with the Welsh test, and am not sure what it means to “look at the significance of the ANOVA test” (line 211-212.)

Discussion
Line 287-290: The authors state that the results of their study suggest that exposure to humans early in life has the potential to be related to animal management and welfare measures, but they do not address how this might be the case.

Line 291-292: The authors mention in the results that they looked for variability with respect to current living environment and did not find one, so it is unclear why here they say that current living environment could be a confound to their results.

Line 304- What do the authors mean by “developmental influence on rearing”? Rearing affects behavioral development; does development influence rearing?

Line 308- duration of exposures to what? This sentence is hard to follow.

Line 319: Is coprophagy seen in wild chimpanzees? Maybe it is not an abnormal behavior, but rather one we find unpleasant.

Line 324- “The first is the potential confound was between” doesn’t make sense; there must be a typo.

Line 326-328: I am not sure what this sentence means.

Line 341-342: I am not sure why there would be “inherently less variation” in CHI over the first four years of life compared to the entire lifespan. At some point, all of these subjects moved to facilities in which they had more conspecific interaction and less human interaction. The longer they are housed in this situation, the more daily CHI scores of 1 they will receive, and the less important the first four years become to their overall lifetime CHI. There could be less variation in CHI over the first four years of life, but there could be more variation. I think it depends on individual circumstances, unless I am missing something.

Line 346-347 is awkward.

Line 355-356: It is unclear why research laboratories was brought up, since it has little to do with this study, and is just speculative.

Line 356-359: I am not sure what the authors mean by “directionality of these factors”. Also, saying that CHI could relate to “cortisol levels’ is somewhat of an oversimplification. Rearing can certainly affect stress levels, but that is not the same as “cortisol levels”. There are many “cortisol levels” one could assess; chronic, baseline, in response to stress, etc.

Line 364- The authors state that “The results of this research suggest that future research should focus on developing the best management strategies for how to care for chimpanzees with a variety of early histories in order to meet their social needs”; how do their results suggest that?. The authors seem to be reaching beyond their data. They examined the effect of CHI on very specific behavioral outcomes. They didn’t look at ability to live in or integrate into a group, or social needs, or welfare per se. They found differences in specific behaviors such as coprophagy and grooming. They found no differences in aggression or prosocial behaviors or time spent alone.

Reviewer 4 ·

Basic reporting

I have no expertise in the general area of the research, so my comments will only pertain to the methodology used, as I was instructed to review this portion by the academic editor.

Experimental design

The methodology used is fairly basic and, as described by the authors, is in line with what is considered to be good practice. The use of MANOVA as an initial step, followed by variable specific ANOVA models is in some sense hypothesis testing overkill, the authors' use of multiple testing adjustments is warranted and shows they did their homework.

Validity of the findings

The findings seem valid, again, I only really reviewed the methodology portion of the manuscript.

---

## Round 0.3 · Minor Revisions

· Academic Editor

Minor Revisions

While you have addressed some of the reviewers’ substantive concerns, there are others that still need attention. There are also persistent problems with the basic reporting of your manuscript that require correction before the manuscript could be accepted for publication. Note: Any references to lines in my feedback refer to lines in the reviewing .pdf file.

Substantively, you have not yet adequately addressed a few of the questions one reviewer raised. These include your meaning of “particularly influential” in Line 84, the “directionality of the factors” in Line 334, and the rationale behind “inherently less variation” in Line 315. Your explanations in the letter are insufficient on these three points, and the manuscript could benefit from a few words to clarify that as well for future readers. You may also want to consider clarifying your use of terms “behavioral development” and “expression of behavior later in life” in the manuscript as well.

There are a few problems with your citations/references:
1. At least one of your in-text citations has not been listed in the reference section (Line 274 cites Rogers & Davenport, 1970). Moreover, some items listed in the reference section are nowhere mentioned in the text of the manuscript. See Lines 370-371, 387-392, 397-404, 406-407, 412-415, 422-424, 431-432, 445-446, 458-460, 469-470, 479-482, and 500-501. Please confirm all citations referenced in the text also appear in the reference list, and remove any items from the reference section that do not appear elsewhere.

2. Please review your citation style in the reference section to ensure it conforms to the PeerJ (instructions for authors are available at https://peerj.com/about/author-instructions/). Journal titles should be spelled out in full. It appears that you have been inconsistent with that and inconsistent with your abbreviations (e.g. Lines 433-436 Anim Welf vs. Anim Welfare).

The grammar problems I noted for you in the previous submission have not been adequately mitigated. These problems generate distraction and considerable confusion for readers throughout the manuscript. In some cases these require readers to reread sentences to decipher your intended meaning and detract from the work you are reporting.

1. Data is a plural word. It is incorrect to write “data was,” which you continue to do. See lines 111, 123, and 140. The correct subject/verb agreement is “data were.”

2. Please be consistent with your use of hyphenated terms. Sometimes “mother reared” is hyphenated and other times not. See Lines 31 and 33. This consistency should apply for “human reared” as well. See Line 42. On Line 24 you introduce these categories with hyphenation, so that would be appropriate for continued use throughout the manuscript.

3. This manuscript contains far too many extremely long sentences (often in excess of 40 words!) that are awkward and hard for readers to follow. These should be broken up into smaller, clearer sentences. A number of these winding and confusing sentences appear in the paragraph located on lines 52-71, but there are others.
a. Line 46-48 is long and winding (~40 words), making it quite confusing.
b. Lines 56-58 contain an equally confusing 46-word sentence with awkward syntax. Consider breaking this up or reorganizing some of the clauses.
c. Lines 59-62 contain a 50-word sentence with awkward phrases.
d. Lines 65-69 could also be broken into two separate sentences. If you retain this as one long sentence, you need to add a comma immediately following (McCann et al., 2007).
e. Lines 198-203 consist of one long sentence. The parenthetical should not be set off by parenthesis. Rather, it deserves its own sentence, grammatically speaking.
f. The intended meaning and emphasis in the sentence on Lines 271-275 is unclear, due partially to grammar and word choice. It states, “previous findings…looking specifically at…” However, findings do not look at anything. Investigators and studies look.
g. Lines 281-285. The semi-colon after “factor” on Line 282 should be a period. The comma after “rearing” on Line 284 should be a semi-colon, and a comma should follow “however.”
h. Lines 318-321 contain a very long sentence in which the odd comma positioning and word order generates considerable confusion. Line 318 should clarify to what the 33 and 21 refers. Does this refer to the “human” and “mixed” categories? Then in Line 319 you have a set of commas around “without any exposure to conspecifics to other groups.” That comma usage signals that those words within the commas all describe the six chimpanzees. It seems awkward that “to the other groups” would be included in that description. Rather, you probably intended to say that despite the small sample size it was still important to draw comparisons among those individuals who lacked exposure to conspecifics. You need to revise to make your point clearer here. In other words, you don’t adequately explain what “this” is, so the second half of that very long sentence (lines 320-321 beginning with “because this would give us the most information…”) is also unclear.
i. The sentence in Lines 335-339 would be clearer if split into two, with a break following “…as evidenced by the wide-ranging standard deviations…” on Line 337 and a new sentence starting with “Future studies…”

4. Commas (excess ones and absent ones alike) continue to wreak havoc in this manuscript.
a. Line 15 should not have a comma after “environment”
b. Line 19 should not have a comma after “human”
c. Line 24 needs a comma before “such as”
d. Line 27 should have a comma after “rocking”
e. Line 28 should not have a comma before “tended”
f. Line 32 should not have a comma after “sanctuaries”
g. Line 40 should not have a comma after “likely”
h. Line 63 should have a comma after “peer raised”
i. Line 75 needs a comma after “(CHI)”
j. The comma in Line 82 after “life” should be replaced with a semi-colon, and a comma should be added after “however”
k. Line 83 needs a comma after “study”
l. Line 96 contains an unnecessary comma after “social groups” where it instead needs the insertion of the word “and”
m. Line 115 needs a comma after “captive-born”
n. Line 134 needs a comma at the end after “sites”
o. Line 176 needs a comma after “multiple corrections”
p. Line 182 needs the commas moved – they belong inside each ending quotation marks (i.e. “human,” “mixed,” and “chimpanzee”).
q. Line 189 needs a comma after “partners”
r. Line 197 needs the clause “which was significant in the original ANOVA test” flanked by commas (i.e. a comma is needed after “test”)
s. Line 257 should not have a comma after life but should have the word “as” inserted
t. Line 269 needs a comma after “housed”
u. Line 303 should not have commas setting off “and management styles”
v. Line 306 should have a comma after “enrichment”
w. Line 310 uses a period after “other” but should use a semi-colon there. However should be made lowercase and followed by a comma. A comma should also then be added after “institutions” on line 311.
x. Line 313 needs a comma after “exposure”
y. Line 313 does not need a comma after “In this analysis”
z. Line 332 needs a comma after “cortisol)”
aa. Line 350 does not need a comma after “chimpanzees”

5. In a few places you end your sentences with hanging prepositions. For example, the sentences in Lines 49-51 and 140-141 end with “for.” While this structure is sometimes accepted in informal spoken language, it is not acceptable for the formalities of scholarly publication.

6. There are a few places where spacing seems off (extra spaces between some words and missing spaces between others) or a word seems missing. This often happens during revisions.
a. Line 97 a space needs to be inserted between the words “cages” and “during”
b. Line 328 needs a word between “life” and “relation” (perhaps “in”?)
9. Noun/pronoun disagreement persists in this manuscript.
a. Line 151 notes “he/she spent...their time.” That should be his/her time.
b. Line 299 notes “a chimpanzee’s…their early history exposure.” That should be his/her exposure.

7. Run-on sentences are a continued problem. This is a common problem when sentences exceed 25 words in length.
a. Lines 289-291(and it also contains extraneous use of hyphens)
b. Lines 300-302

8. Sentence fragments have been introduced by incorrect punctuation. Lines 305-306 incorrectly uses a semi-colon but does not contain a complete thought in the subsequent clause. If you delete “of which,” the sentence structure would be ok.

9. I believe you edited previous Lines 296-297 (now Lines 286-287) to address a reviewer’s questioning of your description of coprophagy as “abnormal.” The revised sentence introduces new confusion. It was your rationale that should have been clarified, not the sentence structure. Setting off “typically categorized as abnormal” by commas is not helpful. Please revise to address the reviewer’s concern.

I am hopeful you are able to improve the writing quality of this manuscript introducing CHI and your findings in this area.

---

## Round 0.4 · accepted · Accept

· Academic Editor

Accept

Thank you for your attention to the requested revisions. The readability of the manuscript is now much improved by the changes you've made, as I hope you agree. There may be a few remaining issues (such as use of parentheses within parentheses and extraneous spaces), but they can be addressed with the production staff.